# Similarity-aware Positive Instance Sampling for Graph Contrastive Pre-training

## Abstract

Graph instance contrastive learning has been proved as an effective task for Graph Neural Network (GNN) pre-training. However, one key issue may seriously impede the representative power in existing works: Positive instances created by current methods often miss crucial information of graphs or even yield illegal instances (such as non-chemically-aware graphs in molecular generation). To remedy this issue, we propose to select positive graph instances directly from existing graphs in the training set, which ultimately maintains the legality and similarity to the target graphs. Our selection is based on certain domain-specific pair-wise similarity measurements as well as sampling from a hierarchical graph encoding similarity relations among graphs. Besides, we develop an adaptive node-level pre-training method to dynamically mask nodes to distribute them evenly in the graph. We conduct extensive experiments on 13 graph classification and node classification benchmark datasets from various domains. The results demonstrate that the GNN models pre-trained by our strategies can outperform those trained-from-scratch models as well as the variants obtained by existing methods.

## 1   Introduction

Pre-training on graph data has received wide interests in recent years, with a large range of insightful works focused on learning universal graph structural patterns lying in different kinds of graph data [25, 15, 40, 28]. For instance, Hu et al. [15] pre-train graph neural networks on molecules and transfer the learned model to molecular graph classification tasks, while Qiu et al. [25] pioneer pre-training on big graphs. Compared with traditional semi-supervised or supervised training methods for graph neural networks [12, 18, 38, 10, 33], pre-training tasks formulate the training objective without the access of training labels, and they empower graph neural networks to be generalized to unseen graphs or nodes with no or minor fine-tuning training cost. How to define proper pre-training tasks comes as the principal and also the most challenging part in graph self-supervised learning.

Among current works, graph instance contrastive learning based pre-training tasks have been proved effective to learn graph strcutrual information [25, 40]. It preforms contrast between positive/negative instance pairs extracted from real graphs observed in the dataset. Though positive pairs for graph contrastive learning seems easy to define for those tasks not performed on graph instances, like DeepWalk [22], node2vec [11], where near node-node pairs are treated as positive pairs and Infomax based models like DGI [34], InfoGraph [31], where node-graph pairs from a same graph are treated as positive pairs, it is not the case for graph instance contrastive learning. Attempts from previous literature mainly focus on devising suitable graph augmentation methods, such as graph sampling [25, 40], node dropping [40], edge perturbation [40], and diffusion graph [13] to get positive graph instances from the original graph. Despite the achievements they have made using such graph data augmentation strategies, we assume that such perturbation based graph data augmentation methods

Submitted to 35th Conference on Neural Information Processing Systems (NeurIPS 2021). Do not distribute.

are not universal strategies to get ideal positive samples preserving *necessary information* for graph contrastive learning for various kinds of graph data such as molecular graphs, social graphs, and academic graphs.

We make our assumptions on the *necessary information* that should be preserved in positive instances in the contrastive learning process, which though have not been proved theoretically, are reasonable and are arrived from the re-thinking of the purpose and inherent principle of the contrastive learning and what should positive samples preserve to get an effective method. Such unproved but reasonable assumptions for positive instances are as follows:

- Positive instances should be semantically similar with the target instance;
- Positive samples for the same target instance should also be similar with each other;
- Positive instances should preserve certain domain information if necessary.

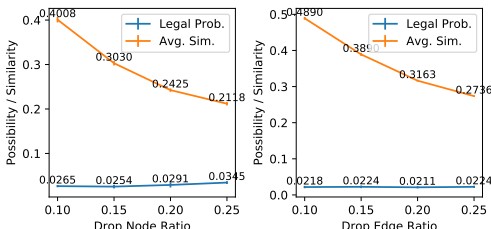

Figure 1: The fingerprint similarity scores (in orange) and percent of legal molecular outputs (in blue) generated by graph data augmentation strategies: *dropping nodes* and *dropping edges* from a same molecule w.r.t. the ratio of nodes / edges being dropped on 1000 molecular graphs. The fingerprint similarity and the percent of legal molecular outputs decreased dramatically even for the small proportion of nod/edge dropping.

Based on such assumptions, we can see that some widely used graph data augmentation strategies cannot always get positive instances with such properties preserved when being applied on different kinds of graph data. As shown in Fig. 1, simple edge-perturbation or node-dropping for molecular graph contrastive learning strategies can hardly get legal graph instances given the fact that molecules are specifically formulated in accordance to strict chemical constraints which will be easily broken if some edges/nodes, even of a very small number, are perturbed. Moreover, subgraph sampling strategy, though effective when applied on graphs without node/edge attributes, may always lead to positive instances that are dissimilar with the target instance when applied on molecular graphs. Statistical results for subgraph sampling and another data augmentation strategy suffering from similar problems – attribute masking, are presented in Appendix A.5.1.

Thus, in this paper, we move beyond the widely used graph data augmentation strategies for an effective and more universal method to get positive graph instances for graph instance contrastive learning. We propose a simple but effective similarity based positive instances sampling strategy that can be applied on various kinds of graph data. Unlike previous methods that construct contrastive pairs by graph augmentation, our method encodes the pair-wise similarity information, measured by certain domain-specific similarity/proximity, into a hierarchical structure and selects positive graph instances from such a structure which ultimately maintains the legality of the sampled instances and high similarity to the target graphs (see Appendix C for details). Moreover, we also propose an improvement for a widely-used node-level pre-training strategy [15], which, together with our similarity aware graph positive sampling strategy, brings us an upper strategy design philosophy. That is, the necessary of introducing prior knowledge or bias in random strategies.

We conduct extensive experiments on three representative kinds of graph data: molecular graphs, social graphs as well as big social and academic graphs where nodes are of interest to demonstrate the effectiveness and superiority of our proposed sampling based strategy over previous graph contrastive learning strategies and also some other strategies not based on contrastive learning for different kinds of graph data. Besides, some additional experiments which try to transfer the GNN models pre-trained on molecular graph dataset to downstream social graph classification task let us have a glimpse of the potential possibility of the pre-trained models' ability to capture universal graph structural information underlying different kinds of graph data as well as the possibility to get such a universally transferable pre-trained model. Similar things have been explored in other domains such as multi-lingual language models. However, to our best knowledge, we are the first to propose such possibility for pre-trained GNN models, which, though lacks further and thorough exploration in the paper, can probably point out a new possibly meaningful research direction and cast light on successive work.

## 2 Related Works

**Graph Representation Learning.** How to generate expressive representation vectors for nodes or graphs that can capture both node-level information, like node attributes and node proximities [32, 22, 11], as well as graph-level information, like structural proximity between nodes [26] and graph property [10], is a vital question and has aroused great interests from graph learning community. Common approaches include unsupervised manners [22, 11, 32, 31, 34, 42, 24, 23], which always adopt a shallow architecture, semi-supervised and supervised approaches [33, 18, 12, 10, 38], which always leverage expressive graph neural networks to capture critical information from both graph structure and node/edge attributes. In this work, we adopt graph neural networks as our graph encoder to generate expressive representations for nodes or graphs.

**Contrastive Learning.** Contrastive learning has proved its efficiency to learn highly expressive representations in Computer Vision domain [5, 14]. Moreover, contrastive learning has also been used in graph learning for a long time, like doing contrast between node-node pairs [22, 11] to encode various node proximities into node representations. Recently, there are also efforts focusing on using contrastive learning on graph instances to learn instance-level representations that can be aware of critical graph structural information [25, 40]. In this work, we also focus on graph instance contrastive learning, but turn to approach this problem in a new manner.

**Graph Pre-training.** Pre-trained models have proved their highly transferable ability when being applied on downstream datasets in other domains, such as the language models [6] in NLP domain. Famous pre-training strategies for GNNs on graph data largely fall into two genres: node-level and graph-level strategies. Node-level strategies aim to design proper tasks that can help GNNs learn node/edge attribute distribution information [15, 28]. More universally, graph-level strategies try to learn design tasks that can learn structural information for both nodes and graphs [25, 40]. In this work, we aim to design more powerful pre-training strategies for graph data from both graph-level and node-level.

## 3 Preliminary

We denote an attributed graph as $G(\mathcal{V}, \mathcal{E}, \mathcal{X})$, where $|\mathcal{V}| = n$ refers to a set of $n$ nodes and $|\mathcal{E}| = m$ refers to a set of $m$ edges. We denote $\boldsymbol{x}_v \in \mathbb{R}^d$ as the initial feature of node $v$ and $\boldsymbol{e}_{uv}$ as the initial feature of edge $(u, v)$.

Graph Neural Networks (GNNs) can be modeled as the a messaging passing process, which involves neighborhood aggregation among nodes in graph and message updating to the next layer. Namely, the general message passing process is defined as:

$$\boldsymbol{m}_v^{(l+1)} = \text{AGGREGATE}(\{(\boldsymbol{h}_v^{(l)}, \boldsymbol{h}_u^{(l)}, \boldsymbol{e}_{uv})|u \in \mathcal{N}_v\}),$$
$$\boldsymbol{h}_v^{l+1} = \sigma(\boldsymbol{W}^{(l)}\boldsymbol{m}_v^{(l+1)} + \boldsymbol{b}^{(l)}),$$

where $\boldsymbol{h}_v^{l+1}$ refers to the hidden state of $v$ at $(l+1)$-th layer with $\boldsymbol{h}_v^{(0)} = \boldsymbol{x}_v$ and $\boldsymbol{m}_v^{(l+1)}$ refers to the aggregated message of $v$ at $(l+1)$-th layer. $\mathcal{N}_v$ denotes the neighbor node set of node $v$. AGGREGATE($\cdot$) aggregates the hidden states of $v$'s neighbor nodes and edges, such as mean/max pooling and graph attention[38, 33]. $\sigma(\cdot)$ is the activation function, such as ReLU($\cdot$). $\boldsymbol{W}^{(l)}$ and $\boldsymbol{b}^{(l)}$ are the trainable parameters. If the model $\mathcal{M}_L$ contains $L$ layers, the output of last layer $\{\boldsymbol{h}_v^{(L)}\}_{v \in v}$ usually represents the node-level embeddings of input graph. Moreover, the graph-level embedding $\boldsymbol{h}_G$ is derived by simply applying a READOUT function as

$$\boldsymbol{h}_G = \text{READOUT}(\{\boldsymbol{h}_v^{(L)}\}_{v \in \mathcal{N}_v}).$$

Representations generated by GNNs over graphs, including node-level and graph-level representations, are meaningful embeddings to perform various downstream graph learning tasks, like node classification [44, 4], graph classification [31, 15, 28], and so on.

## 4 Similarity-aware Positive Graph Instance Sampling

In this section, we propose our similarity-aware hierarchical graph positive instance sampling method to sample positive graph instances with three kinds of information mentioned in Sec. 1 preserved. We

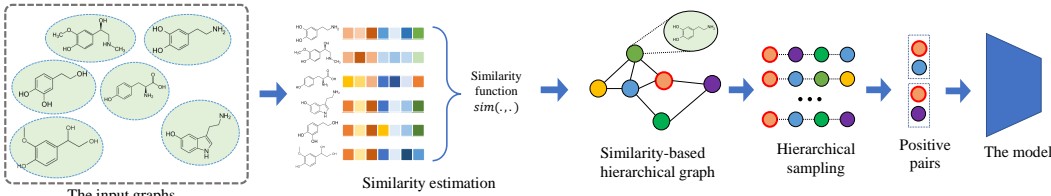

Figure 2: Illustration of the hierarchical graph instance sampling process for the molecular graphs.

first explain our motivation w.r.t. why we turn to other graph instances in the pre-training dataset for positive instances and why it may work for graph data. Then we propose our sampling strategies as well as two versions of the sampling process. We also give some further discussions for such two sampling strategies. Moreover, we also propose an improvement of the widely used node-level pre-training strategy, which is an additional contribution of our work.

## 4.1 Motivation: Sampling or Constructing?

As discussed in Sec. 1, it is hard to design a clean and elegant data augmentation strategy universally for various kinds of graph data to get positive instances that are similar enough with the target graph instance and can also preserve necessary domain specific information. Since what we care about for positive graph instances are their similarity with the target graph instance, rather than the way to obtaining them, we move beyond popular graph data augmentation skills and propose to sample positive instances from the pre-training dataset for the target graph instance. Specifically, we propose to use approximate similarity functions that can reveal the semantic similarity between two graph instances to some extend to estimate the semantic similarity scores between two graph instances. The similarity relations between each pair of graphs are then encoded into a similarity hierarchy, which is then used for positive instance sampling. We also make some further discussions for the proposed similarity-aware sampling process, which may inspire future design for other sampling strategies.

## 4.2 Similarity-aware Positive Graph Instance Sampling

Following [1], we assume that each graph instance $G_i \in \mathcal{G}$ has its semantic class $\text{class}(G_i) = c_i$. Thus, the optimal positive sampling strategy should choose graph instances of the same semantic class with the graph instance $G_i$ as its positive instances. Formally, the rate for sampling graph $G_j$ as the positive instance of $G_i$ is:

$$P_i^+(G_j) = \begin{cases} \frac{1}{|\mathcal{G}_i^+|} & \text{If class}(G_i) = \text{class}(G_j), \\ 0 & \text{otherwise} \end{cases} \tag{1}$$

where $\mathcal{G}_i^+ = \{G_k | G_k \in \mathcal{G}, \text{class}(G_k) = \text{class}(G_i)\}$ is the set of graph instances of the same class with graph instance $G_i$. We can then assume that there exists a ground-truth semantic similarity function $\text{sim}_{\text{gt}}(\cdot, \cdot)$ which reveals whether two graph instances belong to a same semantic class accurately:

$$\text{sim}_{\text{gt}}(G_i, G_j) = \begin{cases} 1 & \text{If class}(G_i) = \text{class}(G_j), \\ 0 & \text{otherwise} \end{cases} \tag{2}$$

However, we have no knowledge of the such ground-truth semantic similarity function since our pre-training graph datasets are always unlabeled. Thus we propose to use approximate similarity functions that can be obtained from the real-world and applied in practice easily to estimate the similarity between two graph instances. We can make some assumptions for the chosen approximate similarity functions to ensure their good quality, which are deferred to Appendix B.1.

Specifically, we choose a similarity score function $\text{sim}(\cdot, \cdot)$ to estimate the semantic similarity between two graphs. To further use the similarity measurement to perform flexible positive sampling, we propose a two-step approach[1] to encode pair-wise similarity into a more abstract and structural hierarchy efficiently – a similarity-based hierarchical graph $\mathcal{H}(\mathcal{G}, \mathcal{E}_H)$, where $\mathcal{G}$ is the set of graphs in our pre-training dataset, $\mathcal{E}_H$ is the edge set. Formally, we introduce a similarity threshold $\tau (0 < \tau <$

---

[1]Please refer to Appendix A.4.1 for details.

1), and based on which the edge set is defined as: $\mathcal{E}_H = \{(G_i, G_j) | sim(G_i, G_j) \geq \tau, G_i \in \mathcal{G}, G_j \in \mathcal{G}\}$. Many similarity functions are good candidates for $sim(\cdot, \cdot)$ such as fingerprint similarity [27] for molecular graphs, Weisfeiler-Lehman Graph Kernel [29] normalized similarity for graphs without node/edge attributes and node proximity for nodes in a big graph.

The constructed hierarchical graph, which encodes more information beyond pair-wise similarity[2], can be used to design flexible sampling strategies for positive graph instance selection. We propose two sampling strategies:

- **First-order neighbourhood sampling.** For each graph $G_i$, sample a one-hop neighbour set of a fixed size as its positive instances.
- **High-order graph sampling.** We perform $l$-hops random walks starting from graph $G_i$ for $k$ times and choose positive instances according to their appearance frequencies.

An illustration for HGC is presented in Fig. 2. We will give some further discussions w.r.t. why we use similarity for positive instance sampling and how would high-order sampling potentially benefit the sampling process and the resulting positive instances in the next section.

### 4.3 Further Discussion for Similarity-aware Sampling Strategy

In this section, we want to answer two questions: **Q1:** Why we still sample positive instances based on approximate pair-wise similarity scores, though it may not be an accurate similarity estimation? **Q2:** How would high-order sampling potentially benefit the sampling process and the resulting positive instances? Moreover, we also propose some further discussions for the proposed similarity-aware positive instance sampling strategy.

To begin with, we propose a property of the contrastive learning that is intuitively correct:

**Property 1.** *Avoiding false-positives is important in the contrastive learning process.*

Here, "false-positives" denotes positive instances selected by a non-optimal positive sampling strategy whose semantic classes are not same with the target graph instance. We explain why such a property holds in Appendix B.2 in detail, though it should be correct intuitively.

Then, **Q1** can be answered by proposing the following property of the positive instances sampled according to their similarity scores with the target graph instance:

**Property 2.** *If the similarity threshold $\tau$ is changing in a proper range, an instance that has a high similarity score with the target instance will also has a high probability to be a ground-truth positive instance.*

We would explain why this property holds in detail in Appendix B.3, based on our assumptions on good properties of the approximate similarity function (Def. 2). Thus, the answer for **Q1** could be: *sampling positive instances according to their similarity scores with the target graph instance may help avoid sampling false-positives.*

To answer **Q2**, we first propose one limitation of the first-order similarity sampling strategy by pointing out a crucial property of the ground-truth similarity function that the approximate similarity functions always fail to preserve – *the transitivity of the ground-truth similarity function*:

**Property 3** (Transitivity of the ground-truth similarity function). *Ground-truth similarity function is transitive: if $sim_{gt}(G_i, G_j) = 1$ and $sim_{gt}(G_i, G_k) = 1$, then $sim_{gt}(G_j, G_k) = 1$.*

Such transitivity of the ground-truth similarity function ensures the transitivity of the relations between nodes in the hierarchical graph constructed based on the ground-truth similarity measurement. However, it is obvious that relations between nodes in our constructed similarity-based hierarchical graph – represented by edges, are not fully-transitive. It is because that the approximate similarity function we use in practice is not an optimal one.

We introduce the definition of connectivity and connectivity order between nodes in the graph in Appendix B.4. The transitivity of the ground-truth similarity function ensures that $G_i$'s positive instances sampled by first-order neighbourhood sampling strategy can have connectivity orders with $G_i$ ranging from 1 to $|\mathcal{G}_i^+| - 1$. It is hard for first-order sampling strategy applied on the hierarchical

---

[2]Such information will be discussed in Sec. 4.3.

graph constructed in practice to get positive instances that also have high-order connectivity (e.g., second-order connectivity) with the target graph instance. The reason is that first-order information cannot reveal high-order information (e.g., high-order connectivity with the target graph instance) in the constructed hierarchical graph, while it can fully reveal higher-order connectivity in the constructed hierarchical graph based on ground-truth similarity function (i.e., if a graph instance $G_j$ is 1-connected to $G_i$, then it is $2, 3, ..., |\mathcal{G}_i^+| - 1$ connected to $G_i$ as well).

We can prove that first-order neighbouring positive instances sampled by second-order sampling process are more likely to be connected with each other (see Appendix B.4 for details). This can remedy the limitation of the first-order sampling strategy, which cannot guarantee the similarity between positive instances. Moreover, it can be empirically verified that positive instances that are both first-order and second-order connected to the target instance are also more similar with the target instance. More details are deferred to Appendix B.4. We also expect that higher-order sampling process can bring more benefit to the resulting positive instances and worth trying in practice.

Additionally, we propose further discussions w.r.t. how would the changing similarity threshold $\tau$ influence the balance between the increasing sampling rate estimation accuracy for ground-truth positive instances and the risk of sampling more false-positive instances. Detailed discussions are deferred to Appendix B.5.

## 4.4 Adaptive Masking for Node-level Pre-training

In this section, we propose our improvement of the widely used *attribute masking* node-level pre-training strategy: Adaptive Masking, which is designed for attributed graphs only. As introduced in [15], *attribute masking task*, which is inspired from "masked language model" (MLM) in NLP, helps the model learn node/edge attribute distribution across the graph. Formally, attribute masking task is defined as:

**Definition 1.** *(Attribute masking task): Given an attributed graph $G(\mathcal{V}, \mathcal{E}, \mathcal{X})$, a target node $v \in \mathcal{V}$ and its corresponding feature vector $\boldsymbol{x}_v$, attribute masking task is first to mask a subset of the features $\boldsymbol{x}_{sub} \subseteq \boldsymbol{x}_v$ in feature vector $\boldsymbol{x}_v$ and produce a new feature vector $\boldsymbol{x}_v'$ for node $v$. Then let a model $\mathcal{M}$ to make the prediction of the masked feature set $\boldsymbol{x}_{sub}$ given the new feature vector $\boldsymbol{x}_v'$ as input.*

Hu et al. [15] follows the same protocol in MLM by uniformly selecting the nodes set from graphs to construct the attribute mask task. But, we argue that the uniform selection may break structural relations among nodes in graphs so that the model may miss critical information for node attribute distribution from such relations. We introduce a toy example in the Appendix A.4.2.

Inspired by Kmean++[2], which aims to obtain the good initial centroids with widely separated in space, we also adopt the adaptive masking (AdaM) to generate the mask node set within less correlations. In particular, we divide the masking process into $T$ steps. At the first step, we uniformly sample a small mask set. Secondly, the masking weight of each candidate node is adaptive by function PScore. The detail of PScore is demonstrate in Algorithm 2 (see Appendix A.4.2). In PScore, for the candidate node $v$, we calculate the similarity of model output between before and after masking. High similarity indicates that node $v$ is not influenced by the mask operation at the current step, resulting in the low correlation between node $v$ and current mask set $S_{\text{cur}}$. Finally, we randomly sample a node set $\mathcal{K}$ with the probability constructed by masking weight. The algorithmic details are provided in the supplementary material.

According to the adaptive masking operation, we can dynamically adjust the importance of nodes during training and obtain a more representative mask node set for the attribute masking task. Such intuition is further discussed in Appendix A.7.

## 5 Experiments

### 5.1 Experimental Configuration

**Pretraining Data Collection.** We conduct the pretraining on four datasets from various domains: 1). *academic and purchasing graphs*: we collect four data sources from Deep Graph Library [36] and merge them into one pretraining dataset dubbed AP_NF. 2). *social graphs*: we construct two pretraining datasets termed SocS_NF and SocL_NF. SocS_NF contains five data sources, while

Table 1: Experimental results (ROC-AUC) on molecular datasets. The numbers in brackets are standard deviations. Numbers in gray are the best results achieved by backbone models. Bold numbers represent the best results by different backbones. Bold numbers in green represent the best results over all backbones.

| Backbone | Strategy | SIDER | ClinTox | BACE | HIV | BBBP | Tox21 | ToxCast |
|---|---|---|---|---|---|---|---|---|
| #Molecules | | 1427 | 1478 | 1513 | 41127 | 2039 | 7831 | 8575 |
| #Prediction tasks | | 27 | 2 | 1 | 1 | 1 | 12 | 617 |
| GIN | GraphCL | $0.5946_{(0.0055)}$ | $0.6592_{(0.0074)}$ | $0.7713_{(0.0057)}$ | $0.7754_{(0.0093)}$ | $0.7050_{(0.0012)}$ | $0.7562_{(0.0024)}$ | $0.6289_{(0.0023)}$ |
| | C_Subgraph | $0.5838_{(0.0022)}$ | $0.6390_{(0.0071)}$ | $0.7736_{(0.0140)}$ | $0.7341_{(0.0079)}$ | $0.6901_{(0.0026)}$ | $0.7521_{(0.0044)}$ | $0.6263_{(0.0061)}$ |
| | Edge_Pred | $0.5949_{(0.0032)}$ | $0.6335_{(0.0168)}$ | $0.7939_{(0.0064)}$ | $0.7757_{(0.0096)}$ | $0.6623_{(0.0229)}$ | $0.7589_{(0.0033)}$ | $0.6456_{(0.0023)}$ |
| | Infomax | $0.5755_{(0.0024)}$ | $0.6944_{(0.0187)}$ | $0.7571_{(0.0094)}$ | $0.7653_{(0.0040)}$ | $0.6929_{(0.0054)}$ | $0.7674_{(0.0020)}$ | $0.6302_{(0.0007)}$ |
| | Attr_Mask | $0.5947_{(0.0083)}$ | $0.6685_{(0.0093)}$ | $0.8064_{(0.0042)}$ | $0.7668_{(0.0106)}$ | $0.6316_{(0.0007)}$ | $0.7657_{(0.0054)}$ | $0.6463_{(0.0029)}$ |
| | Context_Pred | $0.6132_{(0.0050)}$ | $0.6476_{(0.0168)}$ | $0.8055_{(0.0115)}$ | $0.7807_{(0.0054)}$ | $0.7026_{(0.0097)}$ | $0.7715_{(0.0022)}$ | $0.6427_{(0.0024)}$ |
| | HGC | $\mathbf{0.6333}_{(0.0121)}$ | $\mathbf{0.8134}_{(0.0115)}$ | $\mathbf{0.8442}_{(0.0138)}$ | $\mathbf{0.7853}_{(0.0072)}$ | $0.7217_{(0.0042)}$ | $\mathbf{0.7770}_{(0.0022)}$ | $0.6520_{(0.0052)}$ |
| | AdaM | $0.6164_{(0.0051)}$ | $0.7797_{(0.0040)}$ | $0.8224_{(0.0041)}$ | $0.7704_{(0.0073)}$ | $\mathbf{0.7273}_{(0.0146)}$ | $0.7696_{(0.0014)}$ | $\mathbf{0.6603}_{(0.0004)}$ |
| | HGC_AdaM | $0.6183_{(0.0063)}$ | $0.7845_{(0.0499)}$ | $0.8428_{(0.0064)}$ | $0.7839_{(0.0073)}$ | $0.7172_{(0.0052)}$ | $0.7692_{(0.0030)}$ | $0.6537_{(0.0030)}$ |
| GCN | HGC | $\mathbf{0.6243}_{(0.0044)}$ | $\mathbf{0.8638}_{(0.0051)}$ | $\mathbf{0.8405}_{(0.0006)}$ | $0.7724_{(0.0206)}$ | $0.7168_{(0.0014)}$ | $0.7581_{(0.0026)}$ | $0.6490_{(0.0024)}$ |
| | AdaM | $0.6209_{(0.0028)}$ | $0.8553_{(0.0044)}$ | $0.8205_{(0.0120)}$ | $0.7693_{(0.0032)}$ | $0.7018_{(0.0074)}$ | $0.7533_{(0.0059)}$ | $0.6449_{(0.0035)}$ |
| | HGC_AdaM | $0.6164_{(0.0103)}$ | $0.8231_{(0.0325)}$ | $0.8249_{(0.0059)}$ | $\mathbf{0.7946}_{(0.0102)}$ | $\mathbf{0.7189}_{(0.0103)}$ | $\mathbf{0.7636}_{(0.0070)}$ | $\mathbf{0.6525}_{(0.0025)}$ |
| GraphSAGE | HGC | $\mathbf{0.6286}_{(0.0016)}$ | $0.7395_{(0.0284)}$ | $\mathbf{0.8368}_{(0.0008)}$ | $0.7722_{(0.0149)}$ | $0.7129_{(0.0153)}$ | $0.7583_{(0.0012)}$ | $\mathbf{0.6505}_{(0.0004)}$ |
| | AdaM | $0.6148_{(0.0100)}$ | $0.7098_{(0.0244)}$ | $0.8212_{(0.0019)}$ | $\mathbf{0.7730}_{(0.0057)}$ | $0.6982_{(0.0088)}$ | $\mathbf{0.7643}_{(0.0011)}$ | $0.6492_{(0.0004)}$ |
| | HGC_AdaM | $0.6250_{(0.0029)}$ | $\mathbf{0.8127}_{(0.0213)}$ | $0.7812_{(0.0038)}$ | $0.7708_{(0.0053)}$ | $\mathbf{0.7187}_{(0.0019)}$ | $0.7610_{(0.0008)}$ | $0.6442_{(0.0018)}$ |

SocL_NF contains 13 data sources collected from TUDataset [19]. 3). *molecular graphs*: we use the same pretraining dataset with 2 million molecules in [15] and denote it as MolD. The suffix NF indicates "no feature". Since the data sources have different features, we remove all feature and only pretrain these datasets with HGC. The details are presented in Appendix A.1.

**Downstream Tasks.** We mainly evaluate the peformance on two tasks, node classfication and graph classification. For the node classification, we conduct the experiments on two datasets, US-Airport [26] and H-index [41] following the same splitting protocol in [25]. For the graph classification, we conduct the experiments on 11 datasets from molecular graph (7 datasets from [37]) and social graphs (4 datasets from [39]). Details of those datasets are deferred to Appendix A.1.

**Baselines.** For molecular graph classification, we comprehensively compare our pre-training strategies with recent 6 self-supervised learning strategies for graphs. Among them, Edge_Pred, Infomax, Attr_Mask, Context_Pred, are proposed in [15], all of which are node-level pre-training strategies. GraphCL [40] and C_Subgraph [25] are graph level contrastive pre-training strategies. For node classification and social network graph classification, we compare our model with the best result of GCC [25] and several other models (i.e., ProNE [42], GraphWave [7], DGK [39], graph2vec [20], InfoGraph [31], DGCNN [43] and GIN [38]). Details for the implementation, pre-training and fine-tuning settings of baseline models will be discussed in the Appendix A.2 and A.3.

**Pre-training Settings.** We use Adam [17] for optimization with the learning rate of $0.001$, $\beta_1 = 0.9$, $\beta_2 = 0.999$ and weight decay of 0, learning rate warms up over the first $10\%$ steps and then decays linearly. Gradient norm clipping is applied with range $[-1, 1]$. The temperature $\tau$ is set to $0.07$ in HGC pre-training stage. The batch size of MolD pre-training is 256. For SocL_NF and SocS_NF pre-training, the batch size is 32. For the graph classification task, we use mean-pooling to get graph-level representations following [15]. More pre-training details, including backbones, hyper-parameters and training steps are deferred to Appendix A.2.2.

**Fine-tuning Settings.** For each fine-tuning task, we train models for 100 epochs. For graph classification tasks (whether social graphs or molecular graphs), we select the best model by their corresponding validation metrics, while the last model after 100 epochs training on downstream training sets are used for further evaluation on downstream evaluation sets, the same with [25]. We adopt micro F1-score and ROC-AUC as the evaluation measures for different tasks. For molecular dataset, as suggested by [37], we apply three independent randomly initialized runs on each dataset and report the mean and standard deviation. More details are are deferred to Appendix A.2.2.

## 5.2 Results of Downstream Tasks

### 5.2.1 Graph Classification

We evaluate both HGC and AdaM on 7 popular molecular graph classification datasets and HGC on 4 social network graph classification datasets.

**The result of molecular graph classification.** For molecular graph classification datasets, we report our pre-training strategies on different backbones, including GIN [38], GCN [18], GraphSAGE [12]. Meanwhile, since only MolD contain node features, we apply both HGC and AdaM strategies on the molecular datasets. HGC_AdaM indicates the combination of two strategies. As shown in Table 1, we have the following observations: **(1).** GIN model pre-trained by our pre-training strategies can consistently outperform those pre-trained by other existing strategies, with large margin on most of them. The overall absolute improvement is 2.98% in average. **(2).** Specially, HGC can consistently outperform those graph-data-augmentation-based contrastive learning strategies (i.e., GraphCL and C_Subgraph) . It verifies our stand point that the graph data augmentation will lose some crucial domain information and compromise the final performance, while HGC dose not lose such information and leads to better performance. **(3).** Even though GCN/GraphSAGE can not surpass the our pre-trained model on GIN pre-trained model, they still outperform the other pretraining strategy, which reaffirms the effectiveness of our pre-training strategies. **(4).** The combined strategy HGC_AdaM achieve more benefits on GCN and GraphSAGE than that of GIN. We conjecture that GIN encodes the additional noise which is introduced by this simple combination due to its strong expressive power.

Table 2: Results on graph classification datasets. The evaluation metric is micro F1-score.

| Strategy | IMDB-B | IMDB-M | RDT-B | RDT-M |
|---|---|---|---|---|
| # graphs | 1000 | 1500 | 2000 | 5000 |
| # classes | 2 | 3 | 2 | 5 |
| DGK | 0.670 | 0.446 | 0.780 | 0.413 |
| graph2vec | 0.711 | **0.504** | 0.758 | 0.479 |
| InfoGraph | 0.730 | 0.497 | 0.825 | 0.535 |
| DGCNN | 0.700 | 0.478 | - | - |
| GIN(No-Pret.) | 0.734 | 0.433 | 0.885 | 0.635 |
| GIN_GCC (Best) | 0.756 | 0.509 | 0.898 | 0.530 |
| GIN_HGC(SocS_NF) | **0.765** | 0.474 | 0.913 | **0.657** |
| GIN_HGC(SocL_NF) | 0.756 | 0.490 | **0.914** | 0.652 |

**The result of social graph classification.** To check the transferability of HGC, we conduct the finetune experiments on two models pretrained by SocL_NF and SocS_NF. SocL_NF contains the unlabeled data set used in finetune while SocS_NF does not. Table 2 documents the performance of GIN model pre-trained by HGC on SocL_NF and SocS_NF datasets. Such results show that GIN model pre-trained by HGC achieves the best performance on three out of four datasets. The comparison between GIN_HGC and GIN(No-Pret.) also confirms the benefits of HGC. Another interesting observation is that the pretrain model based on SocS_NF can obtain the better performance than than SocL_NF on two out of four datasets. It implies that HGC dose not just memorize the training samples. It can encode the latent structural information from unseen graphs and transfer the knowledge to the downstream tasks.

### 5.2.2 Node Classification.

We evaluate our model pre-trained by HGC on AP_NF on two downstream node classification datasets and summarize the results in Table 3. Among different versions of GCC, the best ones are presented. From Table 3, the model pre-trained by our HGC strategy can outperform the best GCC model on both datasets. It is worth noting that the pre-training dataset AP_NF contains only 70k graphs, which is much smaller than that of GCC(9M graphs). This verifies the efficiency of HGC in the information extraction.

Table 3: Results on node classification datasets. The evaluation metric is micro F1-score.

| Datasets | US-Ariport | H-index |
|---|---|---|
| $|V|$ | 1190 | 5000 |
| $|E|$ | 13599 | 44020 |
| ProNE | 0.623 | 0.691 |
| GraphWave | 0.602 | 0.703 |
| Struc2vec | 0.662 | - |
| GCC (Best) | 0.683 | 0.806 |
| HGC(AP_NF) | **0.706** | **0.824** |

### 5.3 Ablation Study

**How useful are the proposed self-supervised tasks?**

To evaluate the contribution of our pre-training strategies, we compare the the performance of the pre-trained model by HGC and AdaM, with the model without any pre-training, each of which shares the same hyper-parameter setting. Results are summarized in Table 4 for backbone GIN. It can be seen clearly that all GIN models benefit from self-supervised pre-training tasks on all datasets. To be more specific, for GIN, absolute 17.9% ROC-AUC increase is observed on the dataset BACE, 16.5% on ClinTox, and 6.96% on SIDER, leading to 7.53% on average. Furthermore, pre-trained models gain larger improvement on datasets of relatively small size (e.g., BACE, ClinTox and SIDER), which is also observed

Table 4: Effectiveness of the pre-training on GIN. Bold numbers for absolute improvements larger than 0.05.

| | No-Pret. | SS-Pret. | Abs. Imp. |
|---|---|---|---|
| SIDER | 0.5637 | 0.6333 | **+0.0696** |
| ClinTox | 0.6480 | 0.8134 | **+0.1654** |
| BACE | 0.6653 | 0.8442 | **+0.1789** |
| HIV | 0.7475 | 0.7853 | +0.0378 |
| BBBP | 0.6939 | 0.7273 | +0.0334 |
| Tox21 | 0.7580 | 0.7770 | +0.0190 |
| ToxCast | 0.6370 | 0.6603 | +0.0233 |

in [28]. It indicates that self-supervised pre-training helps GNN models learn more inherent graph properties, thus getting better performance in small downstream datasets where labeled graphs are scarce.

**Can we transfer pre-trained models to downstream datasets that are dramatically different from the pre-training one?** It has long been known that the pre-trained model can be generalized to unseen data in pre-training dataset [25, 15, 6, 28, 40]. However, previous literature [25, 15, 40] largely focuses on transferring the pre-trained model to downstream datasets with similar type of data. Here, what we are interested in asking is: can we transfer the pre-trained model to the downstream datasets with clearly different type of graphs compared to the ones in the pre-training dataset? To show this, we demonstrate the case from *molecular graph* to *social network graph* classification. We pre-train GIN in two different ways: one is pretrained by HGC on two social network graph datasets: SocS_NF and SocL_NF, the other is by HGC, AdaM or HGC_AdaM as well as Context_Pred or S_Context_Pred [15] on the molecular dataset MolD.

Table 5: Results for pretraining transferability on graph classification datasets. Numbers in red are the negative transfer cases.

| Pretraining Type | Strategy | IMDB-B | IMDB-M | RDT-B | RDT-M |
|---|---|---|---|---|---|
| None | GIN(No-Pret.) | 0.734 | 0.433 | 0.885 | 0.635 |
| Social | GIN_GCC (best) | 0.756 | **0.509** | 0.898 | 0.530 |
| | HGC(SocS_NF) | 0.765 | 0.474 | 0.913 | 0.657 |
| | HGC(SocL_NF) | 0.756 | 0.490 | **0.914** | 0.652 |
| Molecular | Context_Pred (MolD) | 0.734 | 0.473 | 0.875 | *0.635* |
| | S_Context_Pred (MolD) | 0.763 | 0.460 | 0.818 | 0.625 |
| | HGC(MolD) | **0.768** | 0.504 | 0.912 | 0.656 |
| | AdaM(MolD) | 0.740 | 0.486 | 0.880 | 0.654 |
| | HGC_AdaM(MolD) | 0.713 | 0.509 | 0.896 | 0.665 |

The results are summarized in Table 5, which offers the following observations: **(1).** Perhaps surprisingly, our methods including HGC and HGC_AdaM enable the models pre-trained on molecular graphs to even outperform those pre-trained on social graphs. For example, the accuracy of HGC_AdaM on IMDB-M (0.509) and RDT-M (0.665) is much better than that of HGC(SocS_NF) and HGC(SocL_NF). Apart from the universal graph-level properties, the results also inform that larger pre-training datasets can help the model learn such inherent properties better. **(2).** Different pre-training strategies could deliver different performance. Models pre-trained by graph-level pre-training strategies or combined strategies (i.e., HGC(MolD) and HGC_AdaM(MolD)) can always get better results than those pre-trained by node-level strategies (i.e., AdaM(MolD) and Context_Pred(MolD)), which indicates that graph-level pre-training strategies can help the model learn global graph-level properties that can be easily transferred to other domains. **(3).** We also observe the *negative transfer* brought by the supervised pretraining in some cases. For instance, S_Context_Pred (MolD)[3] get worse performance than its no supervised trained version Context_Pred (MolD) on two datasets: RDT-B and RDT-M. It indicates that simple efforts to learn graph-level properties, such as training with labeled graphs, is probable to be limited in the certain domain, thus performing bad in such cross-domain transfer tasks. Despite this, our HGC and HGC_AdaM still consistently lead to better performance compared to other pretraining strategies, which, once again, versifies our assumption that our proposed graph contrastive learning strategy can learn more universal, even cross-domain, graph-level patterns.

## 6  Conclusion

In this work, we focus on developing an effective, efficient and more universal positive instances sampling method that can be applied on many different kinds of graph data for graph instance contrastive learning. We also propose an improvement for a widely used node-level pre-training strategy to adaptively select nodes to mask for an even distribution (AdaM). Moreover, we also discover the potential cross-domain transferring ability for the pre-trained GNN models. However, there are still some limitations in our work: 1). Though high-order graph sampling can get positive instances of better quality than those obtained by first-order sampling in our analysis, it cannot always outperform the model pre-trained by first-order sampling process. We guess that it is relevant with the pre-training dataset. 2). Just combining HGC and AdaM in a simple manner leads to no significant improvement. Though we make no further investigation into a more effective combination method since it is not the keypoint of the paper, it is a meaningful research direction. 3). We discover the potential cross-domain transferring ability for pre-trained GNN models. It is an interesting point but no further discussion is made in this paper. However, further relevant investigation is interesting and meaningful.

---

[3]Supervised trained by a labeled graph dataset after unsupervised pre-training. See [15] for details.

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
