# OpenReview forum: "Similarity-aware Positive Instance Sampling for Graph Contrastive Pre-training"
_NeurIPS.cc/2021/Conference — NeurIPS 2021 Submitted_

### Official Review · Reviewer_6nrq · 2021-07-16

**Rating:** 5
**Confidence:** 4

**Summary:**

The authors of the paper proposed to select positive graph instances (in graph pre-training) directly from existing graphs
in the training set in order to make them more legitimate.  The proposed scheme is based on pair-wise similarity
measurements as well as sampling from a hierarchical graph encoding similarity relations among graph data sets. An adaptive node-level pre-training method is used to for noe-;evel pretraining tasks as well. Extensive experiments on 13 graph classification and node classification benchmark datasets from various domains are reported to demonstrate the usefulness of the methods.

**Limitations And Societal Impact:**

As mentioned above, the proposed idea is more or less a reuse of similarity-based embedding methods. Although authors claim that it is a contrastive learning framework, it is indeed lacking the important interaction between proximity and similarity relations, therefore its close resemblance with embedding methods such as t-SNE, deep walk makes the novelty of the paper very limited.

**Main Review:**

The authors proposed to use exisent samples in the graph data sets to select positive samples, by carefully designing a similarity-based sampling strategy, in order to improve the resultant pre-training step.  The idea is also used in node-level pre-training tasks. The idea is simple and makes some sense, but there are a numbe rof concerns

(1) There are a number of grammar mistakes and the writing needs to be improved. Sentences are too long to understand quickly and precisely.

(2) Is there any advantage of perturbation-based method for existing method in generating positive samples? For example, is the proposed method for generating positive pairs limited by the size of the graph data set?

(3) The proposed method can be deemed as applying node-level contrastive learning methods (such as deep walk, t-SNE, etc.) in graph-level pre-training tasks; the only difference is that  rather than using proximity relation (such as spatial distance between two nodes in a graph, or co-occurrence of two image patches in a single image), the similarity metric has to be used, since there is no clear prosximity relation defined among graph-level objects. But this is exactly what makes the proposed method very similar to existing embedding methods like t-SNE, and the only difference  is that how to exploit the similarities are defined in more refined manner. Therefore, the novelty of the proposed method is limited. Please note that proximity-based positive/negative sample pairs are very much different from similariy-based ones; in the former case, proximity and similarity can be consistent or conflictive, and they interact with each other; but in the latter case, there is only similarity to be discussed and its scope is therefore not much different than embedding method that relies solely on similarities.


**Time Spent Reviewing:**

3

---

> ### Author Response · Authors · 2021-08-10
> **Response to Reviewer 6nrq (part II)**
>
> **Q2: The proposed method can be deemed as applying node-level contrastive learning methods in graph-level pre-training tasks, leading to limited novelty.**
>
> We agree that sampling from the constructed hierarchical graph for contrastive learning seems like the graph embedding methods which selects nodes via some kinds of proximities for contrast. However, although we all arrive at selecting nodes from a graph to perform contrastive learning finally, the start point, the purpose of contrasting via certain kinds of proximities and the reason of why we arrive at such a solution is not similar. Moreover, the purpose of such two methods are not similar: our work aims to use instance contrastive learning to learn better GNN encoders, but not embed nodes (graph instances) in the constructed hierarchical graph. Despite of the similarity between methods, the problem definition and motivation is totally different from node embedding algorithms. Similarity-aware contrastive pre-training is the method we finally arrive at. The problem it solves better than previous works is more important than the specific methods used to solve the problem. We explain the above mentioned three points as follows:
>
> First, why we arrive at such a solution? We want to argue that although both node embedding algorithms and our work use contrastive learning and random walks, our work is not adapted from such strategies directly. Instead, we arrive at this solution finally. Random walk based node embedding strategies choose to perform various kinds of random walks to transfer nodes in the graph to node sequences. They use random walks (RWs) since RWs can transfer nodes in the graph to node sequences with graph structural information preserved in such sequences. Why we use RWs is because its ability to sample more ideal positive instances than those obtained by using first-order sampling method. Therefore, though the methodology of both our work and node embedding algorithms seems to be similar.  Why we use them is totally different from node embedding algorithms.
>
> Second, the aim of performing graph instance contrastive learning is to learn a transferrable GNN encoder other than embedding graphs in the pre-training dataset. No “function” is learned in embedding algorithms. Thus, the results of our work is a generalizable GNN encoder other than node (graph instance) embedding. Though we both use contrastive learning, our purposes and results are not similar. Graph instance contrastive learning can be seen as an extension of using contrastive learning on GNN self-supervised learning, rather than a reuse of node-level contrastive learning.
>
> Based on the above two points, our work is different from classical node embedding algorithms, even if the solution we arrive at finally is similar with node embedding algorithms. The differences lie in the foundations behind methods and their results.
>
> Third, we sampling positive instances from the hierarchical graph to get higher-quality positive instance than previous works. We adopt the sampling based strategy and finally better solve the problem. Thus, the motivation and the problem we want to solve by using such a method are different from node embedding algorithms.
>
> Moreover, revisiting perturbation based methods to generate positive instances for contrastive learning, we can view them positive instance generation strategy as sampling first-order graph constructed by the augmentation based similarity. Thus, the contrastive learning using such positive instance generation method can also be treated as “applying node-level contrastive learning methods in graph-level pre-training”. The only difference from our work is that they do not construct the “hierarchical graph encoding augmentation based similarity” explicitly. Therefore, we can say that all graph contrastive pre-training strategy are similar with node embedding algorithms, because we can always say that their positive instances are sampled from an implicit / explicit hierarchical graph.
>
> We are not to deny such similarity. In contrast, we have noticed such similarity for a long time. However, it makes us feel interesting since the two problem (graph-level contrastive learning for GNN encoder and node embedding) can be unified into a same framework. The philosophy of science is not trying to become complex, but unified, simple and abstracted, in contrast.

---

> > ### Comment · Reviewer_6nrq · 2021-08-23
> > **Is the hierarchical sampling end-to-end**
> >
> > Thanks for your clarification. I have raised my scores to 5 accordingly, considering that the sampling-based positive graph instance identification does improve the performance, although it is similar to traditional contrastive emedding approaches.
> >
> > In your hierarchical sampling step, you have used 1st or high-order sampling. However, the links (or similarity between graphs) are to be determined by the sim() module, and so how is this sampling implemented exactly? and why do you call it hierarchical?

---

> > > ### Author Response · Authors · 2021-08-24
> > > **Some explanations on the hierarchical sampling process**
> > >
> > > Dear Reviewer,
> > >
> > > Thanks for your recognization of the effectiveness of our proposed method! Below we want to give some explanations on some details of the hierarchical graph sampling process, why we propose the high-order sampling strategy as well as why we call it `hierarchical`.
> > >
> > > The hierarchical graph sampling is a two-stage process, namely big graph construction and hierarchical graph sampling. If we consider the whole process, it is not an end-to-end one. The big graph is constructed at first but only once for each pre-training dataset. After that, many pre-trainings can be conducted based on the big graph.
> > >
> > > In the first stage, we construct a big graph based on all graphs in the pre-training dataset. The construction process can be simply viewed as building edges between those graphs. If the similarity score between two graphs is beyond a threshold, we build an edge between them. The score threshold is selected to make the number of edges about 10 times over the number of nodes in the big graph. Ideally, we should calculate similarity scores between every two graphs to construct the big graph, which is a time-consuming process, however. Thus, in practice, we adopt some heuristic rules to select a set of candidate graphs for each target graph. Thus, the time complexity of the similarity calculation part, the most important part of the big graph construction process, is reduced to be linear to the number of nodes in the big graph. The detailed big graph construction process is explained in A.4.1 Appendix. Detailed time complexity analysis can be found in the `README.md` file under the `supp_code` folder in the supplementary material submitted.
> > >
> > > In the second stage, we sample from the neighborhood around a target graph instance for its positive graph instances which are then used to perform the contrastive learning. This part is processed on the fly during the contrastive pre-training process.  Simply from the methodology aspect, first-order sampling strategy directly samples one graph's first-order neighbors, while high-order sampling strategy samples from its neighborhood for positive instances. It seems that first-order neighbors are those that are the most similar to the target graph. However, there are chances that (1) we may get false-positive instances, and (2) we may fail to sample some instances that are similar to the target instance but not lie in its first-order neighborhood. Such situations are due to the non-ideal similarity score function we use and tricks we use to construct the big graph for speeding up. That's why we introduce high-order sampling strategy. High-order sampling can better capture similarity relations between neighboring graphs, thus leading to more reasonable samples (can be understood as more similar here) than first-order sampling. For example, high-order sampling tends to sample graphs which are both high-order and low-order connected to the target graph. Such graph instances are more likely to be ground-truth positive instances of the target instance.  The detailed analysis w.r.t. the necessarity of sampling positive instances that are similar enough to the target instance, the possible risk of first-order sampling and the potential superiority of high-order sampling are discussed in Section 4.3 and Appendix B.2, B.3, B.4, B.5.
> > >
> > > Why we call it `hierarchical` is mainly because that the more reasonable sampling strategy --- the high-order sampling is aware of the hierarchical similarity relationship between the target instance and its neighboring instances. For example, instances that are both high-order and low-order connected to the target instance are preferred to be sampled. Such `hierarchical` similarity preference helps high-order sampling get instances that are more likely to be ground-truth positive instances.
> > >
> > > Hope that the above can answer your confusion. Please let us know if you have any question about the work.
> > >
> > > Many thanks.

---

> ### Author Response · Authors · 2021-08-10
> **Response to Reviewer 6nrq (part I)**
>
> We really thank the reviewer for pointing out flaws in our writing and pointing out the similarity between our hierarchical graph contrastive learning method and node contrastive methods in graph embedding problems. We give some explanations to address the reviewer's concerns.
>
>
> **Q1: Any advantages of perturbation based methods for generating positive samples? Is the proposed method limited by the size of graph dataset?**
>
> We agree that one of the obvious limitations of the proposed sampling based strategy for positive instances is the limited size of the pre-training dataset, especially for the small dataset. However, we would like to argue that this is not a limitation introduced by the "sampling for positive instances" method itself, but comes from restrictions we add on proper positive instances (i.e., legal graphs preserving domain-specific prior knowledge and similar enough to the target graph instance). It means that the limited available positive candidate pool can be seen as a determinate result to some extend, if we wish that such properties are well preserved in positive instances, regardless of the what specific methods we use (wheter sampling or generating). Moreover, we want to point out that such limitation is not that obvious in practice since pre-training datasets used in practice are always large ones.
>
> We explain the above mentioned two points in detail as follows:
>
> The limitation lies in that we wish all graphs used in contrastive learning are legal graphs with enough similarity to target graphs. Such restrictions are reasonable if we wish the positive instances used in contrastive learning are semantically similar to target graphs, thus resembling ground-truth positive instances. First, enforcing "legality" in positive instances is reasonable since an illegal graph should be classified into “not-a-legal-graph” class, letting alone being semantic similar with the target instance. "legality" limits our possible positive instance candidates to the pre-training graph dataset, if using no extrac knowledge or information. Second, enforcing enough "similarity" is reasonable if the similarity measurement can approximate the semantical similarity to some extend. A straightforward example to support this is an extreme case where we randomly sample other graphs from the pre-training dataset for both positive and negative instances. Clearly, no useful information will be learned by GNN encoders in this contrastive learning setting. The model will even be confused since you tell it one instance is a positive or negative one alternatively. Both sampling based strategy or perturbation based generation strategy can be seen as a process which firstly define a similarity measurement and then sample similar instances based on such measurement. The difference is that we relies on structureal similarity measurements while previous generation strategy treat the perturbation relations as similarity measurements. The unproved assumption is that structural similarity is a better indicator. It is empirically proved by  the performance of contrastve learning pre-trained models using different similarity measurements (see Table 1, 2, 3, 5 for details).
>
> It is just the restrictions we add to the positive instances that improve the performance, and at the same time, limits the size of possible positive instance pool. Sampling based strategy is only the method proposed to satisfy such restrictions. The limitation will still exist even if we change to generation approach, as long as we wish positive instances to satisfy such restrictions. Different strategies would be used to generate graphs with such restrictions satisfied. They will be more complex and not that elegant, we believe.
>
> Moreover, though such limitation do exists, it is not that obvious in practice since he pre-training dataset is always a large one to help the encoder mine enough knowledge underlying data. For example, the molecule pre-training dataset “MolD” used in our work contains 2,000,000 graphs.
>
> We agree that perturbation based strategy can generate much more candidates compared with the sampling strategy. But two concerns of the method make such practical value of such advantage very limited:
>
> The quality of the generated structures cannot be guaranteed. Thus, using perturbation based method to generate positive instances will introduce larger risk to the model being influenced by false-positive instances, thus hard to converge to a proper stage as stated in line 192, Section 4.3 and B.2 in Appendix.
>
> If we can add some constraints to the perturbation process to get high-quality samples (e.g. add restrictions to enforce the generated instances to be legal graphs with enough similarity), the potential positive instance space may be reduced to a even smaller scale than the potential positive sampling space. Moreover, such positive instances generated from a single original graph may lack diversity than choosing from other graphs.

---

### Official Review · Reviewer_qU31 · 2021-07-16

**Rating:** 5
**Confidence:** 4

**Summary:**

In this paper, the authors argue that existing graph unsupervised pre-training models may miss some important information when selecting positive samples, or the reconstructed graphs do not meet some specifications. Based on this motivation, the authors design a similarity-aware sampler to automatically select positive samples from the existing samples for graph contrastive learning. Experiments are also conducted to demonstrate the effectiveness of the method.

**Limitations And Societal Impact:**

The authors elaborate on the article's limitations and social impacts.

**Main Review:**

Strengths: The problem faced by the paper is interesting and timely and the proposed approach seems reasonable. The article is well written, the method is clearly described, and the overall quality is good. The authors also provide the source code to facilitate experimental replication.

Weakness:
1.	Regarding the adaptive masking part, the authors' work is incremental, and there have been many papers on how to do feature augmentation, such as GraphCL[1], GCA[2]. The authors do not experiment with widely used datasets such as Cora, Citeseer, ArXiv, etc. And they did not compare with better baselines for node classification, such as GRACE[3], GCA[2], MVGRL[4], etc. I think this part of the work is shallow and not enough to constitute a contribution. The authors should focus on the main contribution, i.e., graph-level contrastive learning, and need to improve the node-level augmentation scheme.
2.	In the graph classification task, the compared baseline is not sufficient, such as MVGRL[4], gpt-gnn[5] are missing. I hope the authors could add more baselines of graph contrastive learning and test them on some common datasets.
3.	I am concerned whether the similarity-aware positive sample selection will accelerate GNN-based encoder over-smoothing, i.e., similar nodes or graphs will be trained with features that converge excessively and discard their own unique features. In addition, whether selecting positive samples in the same dataset without introducing some perturbation noise would lead to lower generalization performance. The authors experimented with the transfer performance of the model on the graph classification task, though it still did not allay my concerns about the model generalization. I hope there will be more experiments on different downstream tasks and across different domains.
Remarks: 1. The authors seem to have over-compressed the line spacing and abused vspace.
2. Table 5 is collapsed.


[1] Y. You, T. Chen, Y. Sui, T. Chen, Z. Wang, and Y. Shen, “Graph contrastive learning with augmentations,” Advances in Neural Information Processing Systems, vol. 33, 2020.
[2] Y. Zhu, Y. Xu, F. Yu, Q. Liu, S. Wu, and L. Wang, “Graph contrastive learning with adaptive augmentation,” arXiv preprint arXiv:2010.14945, 2020.
[3] Y. Zhu, Y. Xu, F. Yu, Q. Liu, S. Wu, and L. Wang, “Deep graph contrastive representation learning,” arXiv preprint arXiv:2006.04131, 2020.
[4] Hassani, Kaveh, and Amir Hosein Khasahmadi. "Contrastive multi-view representation learning on graphs." International Conference on Machine Learning. PMLR, 2020.
[5] Hu, Ziniu, et al. "Gpt-gnn: Generative pre-training of graph neural networks." Proceedings of the 26th ACM SIGKDD International Conference on Knowledge Discovery & Data Mining. 2020.


**Time Spent Reviewing:**

5

---

> ### Author Response · Authors · 2021-08-10
> **Response to Reviewer qU31**
>
> We thank the reviewer for the recognization for the motivation of the work, pointing out missing baselines in our work as well as proposing their concerns for the similarity-aware positive instance sampling strategy. We give some explanations to address the reviewer's concerns.
>
>
> **Q1: The node-level adaptive masking strategy is incremental: does not discuss existing works related with feature augmentation, lacking results for common node-level classification datasets, lacking better baselines for node-level self-supervised classification task, cannot consitute a contribution, should focus more on graph-level contrastive learning.**
>
> Thanks for point out the limitation of our node-level pre-training strategy and introduce us to these interesting works. We will add the mentioned works to the related work and discuss the similarity (difference) of our work with (from) previous works in detail.
> Notably, our main focus is on formulating the problem of graph-level contrastive learning, proposing our graph-level contrastive learning strategy as well as demonstrating its reasonability and effectiveness through both experiments and comprehensive analysis.
> Similar to the high-quality positive sampling, for the node-level task, we are also interested to selecting high-quality masking candidates, which is also an unexplored topic in the node-level pretraning in graphs. In nut shell, we exploit the idea of K-means++ [1], a classical data mining algorithm, and design a adaptive strategy to sophisticatedly select the candidate masked node based on the current model predictions. These candidate nodes are more representative than that of nodes from uniformly random masking strategy. We also list some statistical results w.r.t. the average distance between masked nodes to reveal its similarity with the “careful seeding” proposed in K-means++.
>
> In this paper, our target is not to surpass all node-level pre-training strategies, but to prove that selecting the quality of masked node is vital for the node making tasks. We test our graph-level pre-training strategy on two node classification datasets to demonstrate that the proposed contrastive pre-training strategy can also be used on graph instances defined on nodes (e.g., subgraphs [2] around target nodes) and thus can be applied to node classification tasks. Moreover, it can also out-perform the previous contrastive pre-training strategy designed for nodes [2], which further shows the effectiveness of the proposed graph-level contrastive learning strategy.
>
>
> **Q2: Compared baselines for graph classification task are not sufficient: missing baseline such as MVGRL [3] and GPT-GNN [2].**
>
> Thanks for introducing us to those interesting papers. We carefully implement the two mentioned baselines (since their code cannot be used directly due to different graph deep learning framework we use) and test them on molecule graphs classification datasets. Pre-training and fine-tuning settings for MVGRL are kept the same with those when pre-training and evaluating our models (i.e., “MolD” pre-training and finetuning settings) due to the large difference of their experimental configurations from ours (e.g. two-stage pre-training and fine-tuning process v.s. one-stage process; GCN backbone in MVGRL v.s. GIN backbone in our work; etc.). Pre-training settings for GPT-GNN are kept to the same with their original settings, except that we use GIN backbone other than HGT used in their original paper.  Fine-tuning settings are kept the same with our models listed in Table 1.
>
> We present their experimental results for GIN backbone on molecule graph classification tasks  as follows:
>
> | Strategy | **sider**      | **clintox**    | **bace**       | **hiv**        | **bbbp**       | Tox21          | Toxcast        |
> | ------------- | -------------- | -------------- | -------------- | -------------- | -------------- | -------------- | -------------- |
> | MVGRL         | 0.5973(0.0057) | 0.6237(0.0169) | 0.7586(0.0065) | 0.7499(0.0030) | 0.6592(0.0013) | 0.7480(0.0040) | 0.6270(0.0073) |
> | GPT-GNN       | 0.5879(0.0033) | 0.6795(0.0084) | 0.7943(0.0082) | 0.7674(0.0078) | 0.6965(0.0146) | 0.7654(0.0066) | 0.6477(0.0043) |
>
>
>
> **Q3: Generalization ability of GNNs pre-trained by contrastive learning strategies using similarity-aware positive instance sampling: whether the proposed similarity-aware positive instance sampling strategy would accelerate GNN-based encoder over-smoothing; more cross-domain experiments are needed.**
>
> “Over-smoothing” is a phenomenon which universally exist in GNNs. A simple and straightforward understanding of the reason of “over-smoothing” is that GNN update each node’s feature by aggregating features from its neighbors, probably leading to indiscriminative features of nodes (even from different classes). However, we can safely assume that it would less likely to appear in GNN encoders trained by contrastive learning. It is because that, contrastive learning pulls similar instances (positive instances and target instances, to be more specific) close to each other while pushes dissimilar instances far from each other. Thus, contrastive learning, regardless of how to get positive samples, will not lead to features of graphs from different classes indiscriminative from each other.
>
> Moreover, what contrastive learning want to do is encouraging similar instances to have similar features. In this work we choose a different similarity measurement by selecting graph instances that are structurally similar to the target instance as its positive instances, different from previous works which perturb the target graph for its similar views.  Why we do not introduce noise into the target graph to construct positive instances lies in that simply adding noise to graph structure and content would lead to illegal and dissimilar graphs. It is because that “structure” is very important for graphs compared with other objects such as images. Even small perturbance cannot guarantee the semantic similarity is kept in the perturbed graphs (see Figure 1, 5, 6, Table 13, 14, 15). Thus, we propose the similarity-aware sampling strategy to get positive instances from the pre-train set. In this manner, the legality can be guaranteed naturally. The similarity can also be preserved by using proper sampling strategy.
>
> As for the generalization ability, we propose two kinds of generalization tests. The first one is regular in-domain generalization test, where the pre-trained model is tested on downstream datasets containing graphs from the same domain with the pre-training dataset. The second one is cross-domain generalization test, where the pre-trained model tested on downstream datasets containing graphs from a domain different from the pre-training dataset. Experimental results on both settings (Table 1, 2, 3 and Table 5) show the superiority of the generalization ability of GNN models pre-trained by our contrastive pre-training strategy to both in-domain and out-domain graphs over previous ones. We think it is enough to empirically prove that GNN models pre-trained by our pre-training strategy have better generalization ability than previous strategies.
>
>
> **Q4: Table 5 is collapsed.**
>
> We apologize for that. We will correct this issue and make the paper look better.
>
>
> **Reference**
>
> [1] Arthur, D., & Vassilvitskii, S. (2006). k-means++: The advantages of careful seeding. Stanford.
>
> [2] Qiu, J., Chen, Q., Dong, Y., Zhang, J., Yang, H., Ding, M., ... & Tang, J. (2020, August). Gcc: Graph contrastive coding for graph neural network pre-training. In Proceedings of the 26th ACM SIGKDD International Conference on Knowledge Discovery & Data Mining (pp. 1150-1160).
>
> [2] Hu, Z., Dong, Y., Wang, K., Chang, K. W., & Sun, Y. (2020, August). Gpt-gnn: Generative pre-training of graph neural networks. In Proceedings of the 26th ACM SIGKDD International Conference on Knowledge Discovery & Data Mining (pp. 1857-1867).
>
> [3] Hassani, K., & Khasahmadi, A. H. (2020, November). Contrastive multi-view representation learning on graphs. In International Conference on Machine Learning (pp. 4116-4126). PMLR.

---

> > ### Comment · Reviewer_qU31 · 2021-08-19
> > **Response on the rebuttal**
> >
> > Thank the authors for the detailed explanation for the questions and novel part of this work. However, these do not dispel my concerns about some of the weaknesses of the paper. I still think the author's contributions at the node and graph level are very confusing and need to be restructured for the paper. Besides, I think the novelty of the proposed method is limited as mentioned by other reviewers. Therefore, I will not revise my score. I hope the author will be able to answer some concerns better and highlight the main contribution in the subsequent versions. Thanks for the work and effort.

---

> > > ### Author Response · Authors · 2021-08-22
> > > **Some explanations on the graph-level contribution and the novelty**
> > >
> > > Dear Reviewer qU31:
> > >
> > > Many thanks for pointing out problems existing in the manuscript. Below we want to give some explanations on the novelty and the graph-level contribution.
> > >
> > > As for the concern of the novelty, we have explained in details in the response to other reviewers, which we would like to highlight as follows:
> > >
> > > - As for the value of this work, we suggest the importance of improving the quality of positive samples in graph contrastive learning algorithms and prove empirically that previous works cannot keep such quality to a satisfactory extend. Thus, to better solve the problem, we propose to sample positive graph instances from existing graphs. Experiments prove that our sampling based strategy can consistently obtain better performance on various downstream datasets than previous generating based strategy. The problem proposal and the adopted approach towards solving it (sampling based strategy, whose inherent properties help it become a better strategy to get high-quality positive samples) are novel parts of the work.
> > > - As for the proposed methods, we arrive at them by informal theoretical analysis mainly on the (1) the necessarity of keeping enough similarity between positive samples and target graphs and (2) the risk of sampling directly from the first-order neighbours and the potential priority of high-order sampling over the first-order samping. Thus, though the proposed first-order sampling and high-order sampling adopt previous sampling methods, why we choose to use them have reasonable foundations. Such simple sampling strategies are already effective enough, which in turn prove the priority of the proposed sampling strategy itself.
> > >
> > > Thus, the contribution of the paper at the graph level mainly lies in (1) we propose the importance of the using high-quality positive samples in graph contrastive learning, which are not lied enough importance on in previous works and (2) we adopt a more reasonable sampling based strategy to get such high-quality samples which further elicits a contrastive learning algorithm that can perform consistently better than previous approaches.
> > >
> > > We hope that above can address your concerns. Please let us know if you have any question about the work.
> > >
> > > Thanks again for pointing out the problems in the manuscript.

---

### Official Review · Reviewer_bttS · 2021-07-16

**Rating:** 6
**Confidence:** 4

**Summary:**

The authors propose a new method for graph pre-training based on instance contrastive learning. They exploit graphs in the training set, to select positive instances that are similar to target graphs. They also propose a new strategy for node-level pre-training.


**Limitations And Societal Impact:**

The limitation and societal (broader) impact are adequately addressed

**Main Review:**

The main contribution is the novel pre-training strategy introduced. The work has potential high impact in the research area, given that the issue of selecting appropriate graphs and datasets for pre-training is still an open and relevant question. The proposed strategy ensured that structurally similar graphs are used for pre-training. An important experimental contribution is also the transfer across domains.

Overall, while technical details are properly provided, I found the general structure of the paper difficult to follow. As a general comment, I would recommend to make the text more smooth and highlight the key steps from sampling to pre-training.

Detailed comments

1) The sampling strategy and concept of the neighbour graph should be better described. The relations with similarity score and specific pre-training strategy should be highlighted clearly.

2) It would also be useful to describe in more details, how the sampling strategy is practically used in pre-training. Maybe a schematic figure would help.

3) I do not see the need of Table 4. The non pre-trained model can be included in table 1. This would also make easier for the reader to validate the effectiveness of the method.

4) The results for no pre-training of table 2 and 3 seem to be missing

---------------------------------------------------------

After authors response

Thank you for your response. I still think the paper needs some restructuring in order to better describe the contribution and improve the clarity of the text. I will keep my score as it is.

**Time Spent Reviewing:**

3

---

> ### Author Response · Authors · 2021-08-10
> **Response to Reviewer bttS**
>
> We are appreciate the reviewer's accepting the value of the problem researched in the paper and potential impact of this work. Below we would like to give some further explanations to answer the reviewer's question as well as address the proposed concerns.
>
>
> **Q1: The sampling strategy and the concept of neighbor graph should be better described. The relations with similarity scores and specific pre-training strategy should be highlighted clearly.**
>
> We explain them as follows: (1) Two sampling strategies are proposed in the paper: the first-order sampling strategy and high-order sampling strategy. As for the first-order sampling strategy, we convert similarity scores to sampling distributions and sample positive instances from such distirbutions. For high-order sampling strategy, we sample neighbours around a node's neighborhood in the constructed hierarchical graph for its positive instances. (2) Similarity scores are used to construct the hierarchical graph for positive instance sampling in the pre-training stage, or are further converted into the sampling distribution for sampling in the pre-training stage. Similarity measurements are chosen according to the characteristic of different datasets. How to choose proper similarity measurements is an interesting question deserving further reserach.
>
>
> **Q2: More details: how the sampling strategy is practically used in pre-training.**
>
> In the pre-training stage, we sample positive instances from the constructed hierarchical graph to perform contrastive learning via the two proposed sampling strategies. In the pre-training stage, we use the two proposed strategies to generate positive instances for a particular graph instance. The detailed sampling process depends on the sampling strategy we use. If we use first-order sampling strategy, we convert the similarity scores between the candidate positive instances and the target instance to a multinomial sampling distribution. We then sample positive instances from this distribution. If we use high-order sampling strategy, we start from the target instance and perform several random walks in its near neighborhood. We can get several graph instance sequences from paths that such random walks pass through. We then calculate the appearance frequency of each graph instance in those sampled sequencies. Such appearance frequencies are then transformed to sampling distribution, from which positive instances are sampled. The transformation process guarantees instances that are more frequently visited in the performed random walks to have higher sampling probabilities. An illustration for the above mentioned high-order sampling process is presented in Figure 2.
>
>
> **Q3: No need of Table 4: the non pre-trained results can be merged to Table 1.**
>
> Its a good suggestion and we will move the non pre-trained results from Table 4 to Table 1. The reason for why we separate the results of non pre-trained model from Table 1 and add Table 4 is that we want to emphasize the benefit brought by our proposed sampling based strategy to GNNs.
>
>
> **Q4: Missing results for non pre-trained models in Table 2 & 3.**
>
> In fact, non pre-trained model's results have already been presented in Table 2. Please kindly check the 8-th row in Table 2 for details.
> The missing results for non pre-trained model in Table 3 are listed as follows:
>
> |	| **US-Airport** | **H-index** |
> |---- |---- | ---- |
> |GIN (No-Pret.)	| 0.640	| 0.761|
>
> At last, we feel sorry if the paper is hard to read through easily due to the bad organizaton, existing grammar errors and long sentences. We will follow the suggestion to further re-organize the paper, correct grammar errors and polish sentences to present our idea as well as the methodology more fluently and clearly.

---

> > ### Comment · Reviewer_bttS · 2021-09-01
> > **Response on the rebuttal**
> >
> > Thank you for your response. I still think the paper needs some restructuring in order to better describe the contribution and improve the clarity of the text. I will keep my score as it is.

---

### Official Review · Reviewer_y4eS · 2021-07-17

**Rating:** 5
**Confidence:** 4

**Summary:**

- The position of this work relates to graph-level contrastive learning. The paper aims to introduce a better way to sample the positive instances, avoiding illegal or poor-quality positive instances.

- The authors propose to select positive graph instances directly from existing graphs in the training set, which can maintain the legality and similarity to the target graphs.

- The positive sample selection is done on domain-specific pair-wise similarity measurements, sampling from a hierarchical graph encoding similarity relations among graphs.

- A side idea using adaptively select nodes to mask for even distribution has been proposed.

- Extensive experiments on various graph classification and node classification benchmark datasets have shown the effectiveness of this simple design’s impact on the final pre-train performance.

**Limitations And Societal Impact:**

- The authors have discussed the limitation of their work to some degree. But the authors fail to discuss one big limitation with is their limitation in terms of the computation complexity. I think the authors should address the trade-off between the performance improvement gain vs. the introduced computation complexity by computing the similarity measurements.

- I do not believe this work has a negative social impact.

**Main Review:**

### Interesting point

This paper mainly aim to discuss the impact of the quality of the positive sample when using contrastive objective to pre-train GNN models. This research angel is interesting and allows the pre-train GNN step to learn more effectively with the quality improved positive samples.

### Main concern

**Methodology aspect**:
- The approach proposed in the paper seems to be a small incremental change on top of the previous GNN pre-train work. The novelty aspect is low.

- A lot of important points only discussed in the appendix. Going back and forth between the main paper and the appendix makes the reader a bit hard to read through the paper.
- Fails to discuss a easy fix to the poor positive sample quality. If we increase the positive sample number, will it be able to match the performance with the proposed high quality (similarity aware) positive instance sampling strategy.

**Experimental result**:

- Fail to discuss the trade off between the performance improvement gain vs. the introduced computation complexity by computing the similarity  measurement between the positive sampled vs. the original graph.






**Time Spent Reviewing:**

4 hours

---

> ### Author Response · Authors · 2021-08-10
> **Response to Reviewer y4eS (part II)**
>
> **Q3: Fail to discuss an easy fix to the poor positive sample quality: will increasing positive sample number help them match the performance of proposed sampling based strategy?**
>
> Thanks for remind us of this possibility. Our main focus is on improving the quality of positive instances used in GCL, since we observed the bad influence that low-quality positive instances would have on the performance of contrastive learned models. The quality of positive and negative instances used in contrastive learning is also discussed in related works from other domains [4]. In contrast, neither the quality nor the number of positive instances is systematically discussed in previous GCL works. We assume the importance of "quality" in contrastive learning with discussions. How the number of positive instances affects the performance of learned model is also an interesting point and worth further discussion.
>
> Thus, we choose the “C_Subgraph” strategy (adapted from GCC [1] actually) list in Table 1 and test the performance of such strategy with more positive samples (i.e., from 1 to 3 and 5). The loss we use is the one proposed in [4] (Eq. 2). Other pre-training and fine-tuning settings are kept the same with those for our models stated in the paper. We list the experimental results as follows:
>
> | **num_samples** | **sider**      | **clintox**    | **bace**       | **hiv**        | **bbbp**       | Tox21          | Toxcast        |
> | --------------- | -------------- | -------------- | -------------- | -------------- | -------------- | -------------- | -------------- |
> | 1               | 0.5838(0.0022) | 0.6390(0.0071) | 0.7736(0.0140) | 0.7341(0.0079) | 0.6901(0.0026) | 0.7521(0.0044) | 0.6263(0.0061) |
> | 3               | 0.5756(0.0039) | 0.6675(0.0143) | 0.7899(0.0135) | 0.7454(0.0059) | 0.7005(0.0034) | 0.7627(0.0048) | 0.6157(0.0031) |
> | 5               | 0.5811(0.0053) | 0.6584(0.0088) | 0.7832(0.0095) | 0.7399(0.0064) | 0.7042(0.0064) | 0.7680(0.0115) | 0.6206(0.0101) |
>
> From the above results, we can see that the overall performance does not change significantly as the number of positive samples increase. It is reasonable since we will always train a lot of epochs in the pre-training stage to fully explore the potential of the model. The performance of a converged model will be more likely to be influenced by the positive instances' quality other than by the number.
>
> The increased number of positive samples may benefit the pre-trained model in some downstream datasets (e.g., improved performance on ClinTox, BACE, HIV, BBBP, Tox21 achieved by the version sampling 3 instances from the version sampling 1 instance). This may be probably due to that some negative influences that low-quality positive instances may have on the model could be cross-counteracted. However, increasing the number of positive instance cannot continuously benefit the pre-trained model if we compare the performance of the model pre-trained by sampling 5 positive instances with 1 or 3 samples version.
>
>
> **Q4: The trade off between the performance gain v.s. the introduced computing complexity.**
>
> We are glad that the computing complexity is pointed out. In fact, our sampling based strategy to get positive instances for contrastive learning will introduce extra time consumption since we need to build a hierarchical graph and sample positive instances from it. However, the introduced time consumption does not count a lot and can be ignored if we mainly care about the efficiency of downstream fine-tuning stage. Moreover, such extra time consumption is worthy compared with the benefit that such sampling strategy brings to the performance of pre-trained models.
>
> Before we go further to discuss the details of the introduced extra time consumption as well as the trade off between such consumption and the improved performance, we would like to remind that we do provide detailed analysis on the extra time consumption of each stage over the whole pipeline in the "README.md" file under the folder "supp_code". We guide the reader to such analysis in Section A.4.3 in the Appendix: “No more than 4 hours for MolD and about 2 hours for SocL_NF, details are presented with code provided (see “README.md” file under the “supp_code” folder.)”.
>
> Below we summary the extra time consumption introduced by the sampling based strategy in the contrastive learning pipeline.
>
> The overall pipeline can be divided into three stages: pre-processing stage (where the hierarchical graph is constructed), pre-training stage (where the constructed hierarchical graph is used for positive instance sampling in the contrastive pre-training process), fine-tuning stage (where the pre-trained model is transferred to downstream tasks. NO extra time consumption is introduced in this stage.). The extra time consumption is mainly composed of the hierarchical graph construction process, and the HGC pre-training stage.
>
> In the pre-processing stage, similarity scores between positive instance candidates and target graph instances are needed to calculate to construct the hierarchical graph. Such calculation process and the graph construction process are relatively efficient. For example, no more than 4 hours are needed in the pre-processing stage for MolD dataset, which contains 2,000,000 graph instances. Fingerprint similarity calculation will take about 3 hours.  As for social network graph datasets, about 1 hour is needed to pre-process the "SocL_NF" dataset, computing on 20 CPU cores in parallel.
>
> In the pre-training stage, performing random walks on the constructed hierarchical graph for positive instance sampling will introduce extra time consumption. Compared with previous contrastive pre-training strategy using data augmentation methods to generate positive instances from the target graph instance such as the subgraph augmentation method [1,2]. Our sampling based method will add about 16.7 minutes to each MolD graph dataset pre-training epoch. Sampling subgraphs as positive instances will take about 40 minutes to train one epoch.
>
> No extra time consumption will be introduced into the fine-tuning stage. Usually, the relative ratio that we go through such three stages for each pre-training dataset is $1 : n : n \times k (n \gg 1, k \gg n)$, where n is the number of pre-training strategies, k is the number of the downstream datasets. Thus, it is the efficiency in the fine-tuning stage that will influence the practical value of our pre-training strategy. Although we will introduce extra time consumption in the pre-processing stage and pre-training stage, the efficiency of our pre-training strategy is not severely affected. The constructed hierarchical graph can be used in each hierarchical graph sampling based pre-training stage and the pre-trained model can be sued in each downstream fine-tuning stage.
>
> The extra time consumption is worthy compared with the improved performance brought by our sampling based contrastive learning methods over previous works. For example, our strategy can bring 15.42% absolute improvement for ClinTox dataset, 7.06% for BACE, 3.86% for SIDER.
>
> **Reference**
>
> [1] Qiu, J., Chen, Q., Dong, Y., Zhang, J., Yang, H., Ding, M., ... & Tang, J. (2020, August). Gcc: Graph contrastive coding for graph neural network pre-training. In Proceedings of the 26th ACM SIGKDD International Conference on Knowledge Discovery & Data Mining (pp. 1150-1160).
>
> [2] You, Y., Chen, T., Sui, Y., Chen, T., Wang, Z., & Shen, Y. (2020). Graph contrastive learning with augmentations. Advances in Neural Information Processing Systems, 33, 5812-5823.
>
> [3] Hassani, K., & Khasahmadi, A. H. (2020, November). Contrastive multi-view representation learning on graphs. In International Conference on Machine Learning (pp. 4116-4126). PMLR.
>
> [4] Khosla, P., Teterwak, P., Wang, C., Sarna, A., Tian, Y., Isola, P., ... & Krishnan, D. (2020). Supervised contrastive learning. arXiv preprint arXiv:2004.11362.

---

> > ### Comment · Reviewer_y4eS · 2021-09-01
> > **Response on the rebuttal**
> >
> > Thanks for detailed discussion from authors' rebuttal. I have read it carefully. I think my question for Q3 and Q4 have been addressed nicely. My concern regarding the limited novelty still hold. I raise my score to 5 accordingly as my final score.

---

> ### Author Response · Authors · 2021-08-10
> **Response to Reviewer y4eS (part I)**
>
> We really thank the reviewer for accepting the motivation of the paper and the potential value it will bring to the research on graph contrastive learning. In the following, we would provide some further explanations to address the reviewer's concerns.
>
> **Q1: The novelty of the paper: it seems to be as small incremental change on the top of the previous GNN pre-train work.**
>
> We agree that what our work focuses on is how to improve the quality of positive instances used in graph contrastive learning (GCL), which seems a relatively small point in GCL.
>
> However, the problem that this work has solved better than previous works is not a trivial one. The quality of positive instances used in GCL is important for the performance of the learned model regardless of its seemingly small value, as suggested by our experimental results (see Table 1 for details). In this work, we have shown that existing data augmentation based strategy cannot guarantee the quality of generated positive instances. In contrast, our work can well preserve good properties in sampled positive instances. Thus, in the view of the real contribution of the work, we solve the problem existing in the quality of positive instances used in contrastive learning to some extend, significantly improving the performance of previous GCL strategies (see Table 1,2,3,5). Thus, our work is of higher value compared with its small change on top of previous architecture.
>
> Moreover, let us move back to the proposed strategy itself, the novelty can be viewed from the comprehensively analyzed reasonability of the proposed strategy. In Section 4.3, B.3, B.4, B.5, we first explain the importance of keeping enough similarity in positive instances with the target instance by analyzing the potential risk that will be brought by dissimilar positive instances. Then, we analyze the potential drawbacks that will be brought by directly sampling positive instances from the most similar ones and propose the high-order sampling strategy which can alliviate such drawbacks to some extend. Therefore, the foundation of the proposed strategy is properly justified. Though simple, the solutions we arrive at are not based on intuitions solely or brought directly from methods proposed in related works in other domains such as image contrastive pre-training. The thinking path of how we design our methods is new and is rarely touched by previous works.
>
> As a last point, the novelty of the work also lies in its problem definition and motivation. What we focus on in this work is the quality of positive instances used in GCL. Most of previous works give little thinking on the quality of such positive instances. We hypothesize that it is curcial to use high-quality positive instances in GCL frameworks. We quantify the term "quality" by specify it to two concepts: "legality" and "similarity". "legality" is specifically introduced for graphs from specific domains where domain prior should be kept such as molecules. It can also be seen as a special  kind of "similarity", actually. Previous works mainly exploit data augmentations techniques to generate positive instances from the target (anchor) instance. Such strategies are mainly adapted from data augmentation methods used in other domains like image contrastive learning. Although they are effective techniques can can generate semantically similar instances for graphs, it cannot be guaranteed that the augmented graphs are semantically similar to original graphs since such data augmentation methods are always destructive and tend to damage graph structure. We verify this point through some statistical experiments on several popular graph data augmentation strategies proposed in previous works (e.g. Figure 1, 5, 6, Table 13, 14, 15). In comparison, our sampling based strategy can do better in preserving enough similarity in positive instances than pervious approaches. Experimental results for the performance of GNN models pre-trained by different GCL strategies on downstream classification tasks demonstrate the superiority of our proposed sampling based strategy. Such results can prove the reasonability of our assumptions on the importance of keeping enough similarity in positive instances used in GCL methods.
>
> **Q2: Bad organization: a lot of important points are only discussed in the Appendix, going back and forth between the main paper and the appendix makes the reader a bit hard to read through the paper.**
>
> We apologize that the paper is not well-organized and the increased difficulty to read through, especially for the methodology section. A lot of important analysis with respect to the the process of how we arrive at the proposed strategies as well as their reasonability are put into Appendix with only brief statements left in Section 4.3. It is because that they are too long to put in the main body due to the limited space. We will re-organize the paper to make it easier to read.

---

### Decision · Program_Chairs · 2021-09-28

**Decision:**

Reject

**Comment:**

The paper proposes to select positive graph instances directly from existing graphs in the training set for constrastive learning, using pair-wise similarity measurements as well as sampling from a hierarchical graph encoding similarity relations. It also develops an adaptive node-level pre-training method. Experimental results show that the GNN models pre-trained by the proposed strategies can outperform baselines.

The reviewers pointed several concerns/aspects for improvement, mainly:
- relative contribution over existing work
- discussion about the baseline of using more positive instances
- discussion on the tradeoff between the gain and the computational overhead
- clarity of presentation
- experiments: more baselines and datasets

The authors have made good responses that significantly strengthened the work. After the response, the consensus is:
- The authors have provided detailed and meaningful discussions on the baseline of using more positive instances, and the tradeoff between the gain and the computational overhead
- The presentation needs improvement
- The authors have provided results of two more baselines that are helpful, while more results will further strengthen the work.
- The novelty/relative contribution of the work remains a concern for the reviewers. Better presentation (importance of the positive samples' quality,relation with existing work) and more thorough and organized experimental evaluation will help illustrate the contribution.



**Consistency Experiment:**

NeurIPS has a long history of experimentation. In 2014, NeurIPS ran an experiment in which 10% of submissions were reviewed by two independent committees to quantify the randomness in the review process. This year, we repeated a variant of this experiment to see how the quality of the review process has changed over time.  This paper was part of the experiment and was therefore assigned to two committees (consisting of reviewers, an Area Chair, and a Senior Area Chair) that reached independent decisions.  If both committees made the same recommendation, this recommendation was followed. If a single committee recommended acceptance, the paper was accepted (with the exception of a few cases in which the other committee identified what we considered a fatal flaw, e.g., an error in a key result).

Both committees reached the same decision: **Reject**

The other committee assigned to the paper recommended **Reject**.  You can find the other set of reviews, along with any follow up discussion with the authors here:
https://openreview.net/forum?id=se6fousHE-